# Antenatal Corticosteroids to Asian Women Prior to Elective Cesarean Section at Early Term and Effects on Neonatal Respiratory Outcomes

**DOI:** 10.3390/ijerph19095201

**Published:** 2022-04-25

**Authors:** Noorazizah Arsad, Nurlina Abd Razak, Mohd Hashim Omar, Mohamad Nasir Shafiee, Aida Kalok, Fook Choe Cheah, Pei Shan Lim

**Affiliations:** 1Department of Obstetrics and Gynecology, Faculty of Medicine, Universiti Kebangsaan Malaysia, Kuala Lumpur 56000, Malaysia; azizah.arsad@gmail.com (N.A.); nurlinarazak@gmail.com (N.A.R.); mhashim@ppukm.ukm.edu.my (M.H.O.); nasirshafiee@hotmail.com (M.N.S.); aidahani.mohdkalok@ppukm.ukm.edu.my (A.K.); 2Department of Pediatrics, Faculty of Medicine, Universiti Kebangsaan Malaysia, Kuala Lumpur 56000, Malaysia; cheahfc@ppukm.ukm.edu.my

**Keywords:** antenatal corticosteroids, elective cesarean section, neonatal respiratory morbidities, transient tachypnea of newborn, NICU

## Abstract

This exploratory study aimed to evaluate the effects of antenatal corticosteroids in singleton pregnancies of Asian women prior to elective cesarean section (CS) at early term on neonatal respiratory outcomes. Methods: This is a pilot and pragmatic randomized trial conducted at a university hospital in Malaysia. Women with singleton pregnancies planned for elective CS between 37^+0^ and 38^+6^ weeks gestation were randomly allocated into the intervention group, where they received two doses of IM dexamethasone 12 mg of 12 h apart, 24 h prior to surgery OR into the standard care, control group, and both groups received the normal routine antenatal care. The primary outcome measures were neonatal respiratory illnesses, NICU admission and length of stay. Results: A total of 189 patients were recruited, 93 women in the intervention group and 96 as controls. Between the steroid and control groups, the mean gestation at CS was similar, 266.1 ± 3.2 days (38 weeks) vs. 265.8 ± 4.0 days (37^+6^ weeks), *p* = 0.53. The mean birthweight of infants was 3.06 ± 0.41 kg vs. 3.04 ± 0.37 kg, *p* = 0.71. Infants with respiratory morbidities were primarily due to transient tachypnea of newborn (9.7% vs. 6.3%), and congenital pneumonia (1.1% vs. 3.1%) but none had respiratory distress syndrome. Only four infants required NICU admission (2.2% vs. 3.1%, *p* = 0.63). Their average length of stay was not statistically different; 3.5 ± 2.1 days vs. 5.7 ± 1.5 days, *p* = 0.27. Conclusions: Elective CS at early term before 39 weeks was associated with a modest overall incidence of neonatal respiratory illness (10.1%) in this Asian population. Antenatal dexamethasone did not diminish infants needing respiratory support, NICU admission and length of stay.

## 1. Introduction

Cesarean delivery prior to onset of labor at term is a recognized risk factor for iatrogenic neonatal respiratory morbidities. Compared to vaginal deliveries, the newborns have higher admission rates to the neonatal intensive care unit (NICU) for pulmonary disorders such as transient tachypnea of newborn (TTN) and respiratory distress syndrome (RDS) [1]. The rising number of cesarean deliveries in the last decade is alarming. National data from 150 countries between 1990 and 2014 showed that the overall cesarean section (CS) rate was at 18.6% [2]. This is much higher than the ideal rate of 15% stated by the World Health Organization (WHO) [3]. In South East Asia particularly, the CS rate was 19.1% in Malaysia, 22.7% in Philippines and 29.6% in Indonesia [4]. Therefore, it is anticipated that the associated complications resulting from this are also expected to increase.

The pathophysiology for neonatal respiratory distress conditions in planned CS at term is slightly different from that of preterm delivery. Absence of labor stress in a planned CS suppresses physiological ‘catecholamine surge’ which serves as a stimulus for fetal lung maturation by enhancing surfactant release [5,6]. Moreover, fetuses are found to have larger residual fluid volume in the lungs due to the absence of vaginal squeeze during the labor process [7,8]. A more important mechanism is the lung epithelial sodium channels are underactive, thus limiting alveolar fluid drainage [9]. The changes required for successful respiratory transition at birth involve the coordinated clearance of fetal lung fluid and surfactant secretion towards maintaining the correct alveolar surface tension and pulmonary compliance in optimizing gaseous exchange. 

Antenatal administration of corticosteroids accelerates the maturation of type II pneumocytes which produces surfactant. It was also hypothesized that the steroids upregulate the number and function of the epithelial sodium channel that facilitates more alveolar fluid to be absorbed after birth [8,9]. Furthermore, it promotes the fetal responsiveness to catecholamine and thyroid hormones that may help accelerate neonatal respiratory transitions at birth [6].

A Cochrane review published in 2018, looking at four RCTs of antenatal corticosteroids administered 48 h before planned CS at ≥37 weeks’ gestation, found a reduction in neonatal respiratory morbidity compared with placebo or no treatment [10]. The incidence of TTN was 2.3% vs. 5.4% (RR 0.43, 95% CI 0.29–0.65), RDS was 2.6% versus 5.4% (RR 0.48, 95% CI 0.27–0.87) whereas admission to the NICU for respiratory morbidity was 2.3% versus 5.1% (RR 0.45, 95% CI 0.22–0.90). However, with only four trials being included and two of them from the same hospital, as mentioned by the author, this would limit the generalizability of the results.

Moreover, the South Asians may be less affected by neonatal respiratory morbidities following CS as more of them were found to deliver spontaneously before 39 weeks than whites (28.2%, 95% CI 27.8–28.6 vs. 16.9%, CI 16.8–17.1) [11,12]. In addition, the incidence of TTN and RDS was lowest at 38 weeks’ gestation whereas in whites, it was lowest at 40 weeks’ gestation. A more recent study in Iran also reported no difference in RDS with or without antenatal steroids [13].

Therefore, delaying the CS until 39 weeks might not be appropriate for the South Asian ethnic group as these women are more likely to progress into labor before the scheduled surgery date. Malaysia has a significant population of people of South Asian descent and we speculate that other Asians (Malays and Chinese that make up the majority of our population) may also possess similar traits. It then raises the question of the need for antenatal corticosteroids for planned term CS in this population balanced against the risk of the fetus developing neonatal respiratory morbidities at 38 weeks and reports of neonatal hypoglycemia and concerns with longer term neurodevelopmental outcomes [12,14,15]. 

This study’s main objective was to evaluate the fetal outcomes following antenatal dexamethasone intervention to elective CS at early term before 39 weeks. Specifically, we would like to compare the incidence of respiratory morbidities such as TTN and RDS between dexamethasone and the no steroid groups, as well as assessing the difference in the rate of NICU admissions, the need for respiratory support or mechanical ventilation and the length of hospital stay. This exploratory pilot trial on the use of antenatal steroids in a population of Asian women that would normally undergo planned CS at early term will help in policy making and future randomized trials. 

## 2. Clinical Study and Methods

This was a randomized pragmatic trial conducted over a 12-month period between December 2016 and December 2017. All patients who were eligible and recruited received antenatal care in the clinics of the Department of Obstetrics and Gynecology, Universiti Kebangsaan Malaysia (UKM) Medical Centre and were scheduled for elective CS between 37^+0^ weeks to 38^+6^ weeks’ gestation. 

All patients have their gestational age confirmed via a first trimester’s dating scan. We excluded women with gestational or pre-existing diabetes mellitus and severe maternal hypertension, suspected intrauterine infections and fetus with major congenital anomalies or intrauterine growth restrictions. Other exclusion criteria included women who had hypersensitivities to dexamethasone or if the women came in labor prior to their scheduled elective CS date.

During enrollment, each patient received a written and verbal explanation regarding the study. A written consent was obtained once they have agreed. This study was approved by the Research Ethics Committee of UKM (JEP-2016-578). The randomization sequence, either to the treatment group or the standard care, no steroid group (control), was generated using a computerized randomization program in blocks of two. Although randomized, the sampling was purposive and not powered for the primary outcome. Pregnant women in the study group received two doses of intramuscular dexamethasone 12 mg, 12 h apart, within 24 h prior to CS. This regime is similar to that used for women presenting with preterm labor in this hospital. The control group received the usual antenatal care without corticosteroids.

Cesarean sections were performed under regional anesthesia (either spinal or epidural) following the hospital protocol. Patients were excluded if desaturation occurred intraoperatively or if there was conversion to general anesthesia. Upon delivery, the babies were attended immediately by the neonatal team in the operating theatre. The neonatal team were not informed regarding the status of antenatal dexamethasone until full assessment was completed. The details of the resuscitation were documented. Neonatal assessments included Apgar scores at 1 and 5 min of life, oxygen saturation at 5 and 10 min and cord blood pH (venous and arterial). All newborns had routine cord blood pH (venous and arterial) assessed. Neonates that were admitted to NICU were followed up in their progress during their stay. 

All data were recorded and analyzed using the Statistical Package for Social Science (SPSS Version 24, IBM, Armonk, NY, USA). For categorical data, results were analyzed using Chi-Square test or Fisher’s exact test, whichever appropriate. For numerical data, the normality was assessed using Kolmogorov–Smirnov analysis. Normally distributed data were then expressed as mean (SD) and assessed using independent Student t-test. Non-normally distributed variables were expressed as median (IQR) and assessed using the Mann–Whitney test. Statistical significance was defined as *p* value of less than 0.05.

## 3. Results

Out of 457 pregnant women scheduled for elective CS, 203 fulfilled the inclusion criteria and agreed to participate in the study. They were randomized into dexamethasone and control groups. However, 14 women were excluded from analysis; 6 from the dexamethasone group and 8 from the control group (Figure 1). Six of them had conversion from regional to general anesthesia, three women came in labor prior to elective dates, two cases were cancelled from the elective listing as the fetuses spontaneously turned cephalic from breech presentation whereas another case was rescheduled for a CS after thirty-nine weeks by the managing consultant. One patient had an emergency CS for fetal distress on admission and another woman delivered at another center. As a result, the total number of women available for this were 93 in the intervention group and 96 women in the control group. 

In general, the demographic characteristics for both study groups were comparable (Table 1). There was no significant difference in terms of the women’s age, ethnicity and BMI at delivery. Seventy percent of participants in both groups were over thirty years old and multiparous, with predominantly Malay followed by Chinese women. More than half of the women were overweight and obese at the time of delivery (58.1% in the dexamethasone group vs. 52.6% in control), with the mean BMI of 27.1 kg/m^2^ and 25.8 kg/m^2^, respectively. Majority of the elective cases were scheduled between 38^+0^ and 38^+6^ weeks. Although there were more women in the dexamethasone group delivering between 37^+0^ and 37^+6^ weeks (39.8% vs. 28.1%, *p* = 0.09), the mean gestational age at delivery was not significant among the two groups (266.1 ± 3.3 days vs. 265.8 ± 4.0 days, *p* = 0.53). 

The neonatal characteristics are as shown in Table 2. Majority of the babies were delivered at 38 weeks rather than 37 weeks’ gestation in both arms and all were born with clear liquor except five cases. The mean birth weight was similar in both groups (3.06 ± 0.41 kg vs. 3.04 ± 0.37 kg, *p* = 0.71).

Table 3 summarizes the neonatal outcomes for both groups in this study. There were no significant differences in the Apgar score, oxygen saturations and cord blood pH between the two groups.

Overall, a total of 19 (10.1%) neonates had respiratory distress. The incidence was similar between the dexamethasone group and the control group (10.8% vs. 9.4%, *p* = 0.75). Fifteen out of nineteen had a final diagnosis of TTN and recovered rapidly with a short duration of nasal cannula oxygen or continuous positive pressure ventilation (cPAP) in the observation room, not requiring admission; nine from the dexamethasone group and six from the control group. None of the newborns had RDS. Another four neonates were suspected to have congenital pneumonia; one in the study group and three in the control group. They were admitted to the NICU for further assessment in view of the need for antimicrobial therapy. No infants required mechanical ventilation or died.

There was no difference in the rate of neonates exposed to antenatal corticosteroids who required non-invasive ventilation (6.5% vs. 7.3%, *p* = 0.70) or NICU admissions (2.2% vs. 3.1%, *p* = 0.63), compared to the control group.

## 4. Discussion

The use of antenatal corticosteroids to reduce neonatal respiratory morbidities in elective CS before 39 weeks’ gestation has been well documented since the publication of the ASTECS trial; which demonstrated its efficacy in reducing the incidence of respiratory distress by 50%, mainly by reducing TTN [16,17,18,19]. This sentinel trial has led to a change in obstetric practice in delaying planned CS to 39 weeks or the use of a single course of antenatal corticosteroids for earlier delivery between 37^+0^ and 38^+6^ weeks’ gestation. Nevertheless, since South Asians may deliver earlier and are known to be predisposed to meconium-stained amniotic fluid (MSAF) at term more than white Europeans, delaying delivery to 39 weeks plausibly exposes them to higher maternal and fetal morbidities [12]. Changing the practice to delay planned CS may render a higher number of emergency CS which predisposes pregnant mother to potential maternal gastric content aspiration, hemorrhage, surgical site injuries and anesthetic complications while putting the newborn at higher risk of meconium aspiration syndrome.

In this study, the race distribution of participants in both groups reflects Malaysia’s ethnic group composition, which predominantly consists of the Malays (69%), followed by Chinese (23%), Indians (6.9%) and other races (1.0%) [20]. The mean age of women in both groups (33.7 ± 4.3 vs. 33.6 ± 4.0 years old) also corresponds to the latest national statistics, reporting that women aged 30–34 years old had the highest fertility rate in the country [21]. The most common indications for planned CS were cesarean in previous delivery as well as malpresentation, which are also in agreement with recent regional data [4].

Our results in this study showed a modest incidence of neonatal respiratory morbidity (10.1%) compared to previously reported observations, [16,19,22] further supporting the fact that Asian newborns may perhaps be less affected by neonatal respiratory morbidities. The neonatal Apgar Score at 1 and 5 min, oxygen saturation at 5 and 10 min and cord blood pH were comparable in both arms. Only 1 in 10 infants developed some form of respiratory difficulties, predominantly mild TTN which resolved with brief oxygen or CPAP support without needing NICU admission. It is arguable if fetal exposure to antenatal steroids is warranted for elective CS prior to 39 weeks against if at all additional benefits to modest and brief periods of support for the respiratory morbidities observed. Similarly, the use of antenatal steroids may be more clinically beneficial in reducing resuscitation and in stabilizing the respiratory condition in the labor room for moderate or late preterm pregnancies between 34 and 37 weeks’ gestation [23,24].

However, it is concerning that antenatal steroids for more mature fetuses have been associated with higher incidence of neonatal hypoglycemia [14,25]. Recently published systemic review and meta-analysis further alerts us to the potential long-term effects on neurodevelopment with antenatal exposure of corticosteroids. In this review, a total of 30 studies were included where the authors highlighted the risk of associated mental, behavioral and neurocognitive disorders in preterm birth with antenatal corticosteroids exposure. The postulated explanation was that fetuses near term are exposed to intrinsic increment of cortisol from self and also from the maternal side. The dose of the injected exogenous corticosteroids is supraphysiological which may interfere in the brain developmental programming as well as the hypothalamic–pituitary–adrenal axis [26].

Our study did not show any significant difference in the outcomes of respiratory disorders in both groups. The majority of the infants were diagnosed to have TTN (9.7% vs. 6.3%), and a few were admitted for work-up as congenital pneumonia (1.1% vs. 3.1%) in the intervention and control groups, respectively, but none developed RDS. We speculate that our findings, consistent with previous studies, suggest that Asian ethnicity may be less affected by RDS compared to whites and Hispanics at early term, possibly from an innate biologically timed earlier onset of labor [11,27]. Earlier maturity of Asian infants may also be associated with a higher incidence of MSAF compared to the whites and a lower incidence of RDS at early term in our population [11,28,29]. Hence, the role of dexamethasone to reduce neonatal respiratory disorder in elective CS at term may confer less benefit to our population, and needs to be balanced against the reported side effects and potential adverse outcomes; these by themselves need larger trials in the future to confirm.

Overall, our NICU admission rate for respiratory morbidities was 2.64% whereas the reported NICU admission rate for the same indication in ASTECS trial was 3.71%, where about half of their study populations were delivered at 39 weeks and above. In contrast, all CS were conducted between 37 weeks and 38^+6^ weeks, but our NICU admission rate remains slightly lower than the ASTECS trial, albeit our study population was significantly smaller. However, it begs the question: What is the role of antenatal corticosteroids in elective CS cases before 39 weeks’ gestation and if it is applicable to the Asian population? Moreover, administration of dexamethasone did not decrease significantly the number of admissions to NICU (2.2% vs. 3.1%, *p* = 0.63) and the length of stay (3.5 ± 2.1 vs. 5.7 ± 1.5, *p* = 0.27), in which the duration was determined by the need for antibiotic therapy in four of five infants who had to be admitted rather than due to the severity of the respiratory condition requiring oxygen or ventilator support. Three infants were treated for early onset pneumonia and another in the control group was treated for mild meconium aspiration pneumonitis.

We did not do a cost–benefit analysis in this study, but we believe this aspect should also be explored further if administration of dexamethasone is cost effective in comparison to the need for transient respiratory support in labor room, NICU admission, medical treatment and length of hospital stay.

The major limitation in this study is that it was a small, single center pilot study with purposive sampling, and although randomized, was not powered for the outcomes. Neither the managing obstetricians or study participants were blinded in this trial. The elective CS procedure was under regional and not general anesthesia, which may not reflect wholly the true potential risks of neonatal morbidities observed in other studies. However, as the neonatal team was not informed regarding the antenatal corticosteroid status at the time of assessment upon delivery, this limited the bias in diagnosing respiratory morbidities and the need for support or NICU admission.

## 5. Conclusions

In summary, our study showed 1 in 10 early term infants developed neonatal respiratory disorder, predominantly mild TTN when elective CS was performed at 37–38 completed weeks. One in four of these infants required NICU admission. Exposure to antenatal corticosteroids did not significantly affect the outcomes. Given the modest risk of mild neonatal TTN when delivered between 37 and 39 weeks in Asian ethnic groups with a presumably shorter gestation to delivery, further trials are needed in studying if the role and impact of antenatal steroids for elective CS before 39 weeks is significant in these populations.

## Figures and Tables

**Figure 1 ijerph-19-05201-f001:**
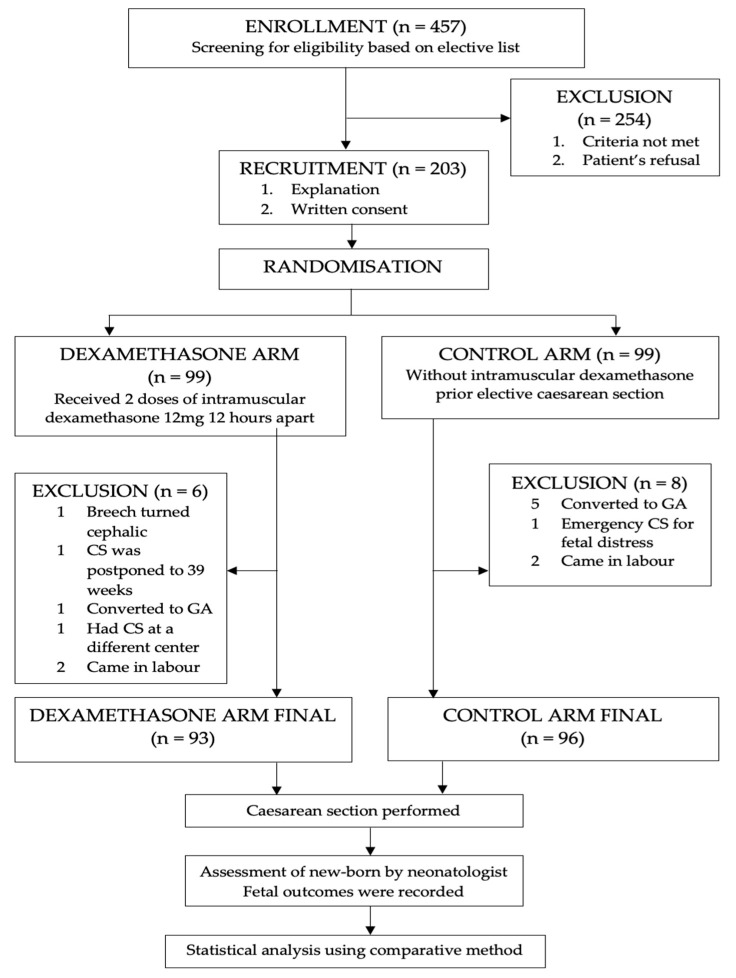
Study Procedure Flowchart.

**Table 1 ijerph-19-05201-t001:** Demographic Data.

Characteristics	Dexamethasone Group		Control Group		*p*-Value
	n = 93		n = 96		
	n (%)	Mean ± SD	n (%)	Mean ± SD	
Age (years)		33.7 ± 4.3		33.6 ± 4.0	0.78 ^a^
Age group					0.52 ^b^
<30	14 (9.8)		19 (15.1)		
30–39	70 (74.0)		71 (75.3)		
≥40	9 (6.3)		6 (9.7)		
Ethnicity					0.22 ^c^
Malay	66 (71.0)		71 (74.0)		
Chinese	22 (23.7)		17 (17.7)		
Indian	3 (3.2)		1 (1.0)		
Others	2 (2.2)		7 (7.3)		
BMI (kg/m^2^)		27.1 ± 5.8		25.8 ± 5.2	0.10 ^a^
BMI group					0.15 ^c^
Underweight	3 (3.2)		4 (4.2)		
Normal	36 (38.7)		41 (43.2)		
Overweight	26 (28.0)		36 (37.9)		
Obese class 1	20 (21.5)		8 (8.4)		
Obese class 2	4 (4.3)		4 (4.2)		
Morbid obesity	4 (4.3)		1 (2.1)		
Gestational age (days)		266.1 ± 3.2		265.8 ± 4.0	0.53 ^a^
Gestational age (weeks)					0.09 ^b^
37^+0^–37^+6^	37 (39.8)		27 (28.1)		
38^+0^–38^+6^	56 (60.2)		69 (71.9)		
Parity		1.9 ± 1.1		1.6 ± 1.2	0.05 ^a^
Parity					0.05 ^b^
Nulliparous	9 (9.7)		19 (19.8)		
Multiparous	84 (90.3)		77 (70.2)		
Indication of CS					0.26 ^c^
Previous CS	62 (66.7)		54 (56.3)		
Malpresentation	12 (12.9)		18 (18.8)		
Placenta Previa	8 (8.6)		5 (5.2)		
CPD	3 (3.2)		3 (3.1)		
Previous hysterotomy	2 (2.2)		1 (1.0)		
or myomectomy					
Others	6 (6.5)		15 (15.6)		

^a^ Independent *t*-test, ^b^ Chi-Square test, ^c^ Fisher’s exact test.

**Table 2 ijerph-19-05201-t002:** Neonatal Characteristics at the time of delivery.

Characteristics	Dexamethasone Group		Control Group		*p*-Value
	n = 93		n = 96		
Gestation, *n* (%)					0.09 ^a^
37 weeks	37 (39.8)		27 (28.1)		
38 weeks	56 (60.2)		69 (71.9)		
Liquor, *n* (%)					0.36 ^b^
Clear	91 (98.9)		88 (95.7)		
Light meconium	1 (1.1)		3 (3.3)		
Moderate meconium	0 (0.0)		1 (1.1)		
Gender, *n* (%)					0.13 ^a^
Male	45 (48.4)		57 (59.4)		
Female	48 (51.6)		39 (40.6)		
Birth Weight (kg), mean (SD)		3.06 ± 0.41		3.04 ± 0.37	0.71 ^c^

^a^ Chi-square test, ^b^ Fisher’s exact test, ^c^ Independent *t*-test.

**Table 3 ijerph-19-05201-t003:** Neonatal outcomes for dexamethasone versus no steroid (control) groups.

Character	Dexamethasone Arm			Control Arm			
	n = 93			n = 96			
	n (%)	Median (IQR)	Mean (SD)	n (%)	Median (IQR)	Mean (SD)	*p*-Value
Apgar score							
1 min		9 (8–9)			9 (9)		0.66 ^a^
5 min		10 (9.5–10)			10 (10)		1.00 ^a^
SpO2							
5 min		93 (89.5–97)			94 (90–96)		0.79 ^a^
10 min		98 (97–99)			98 (96–99)		0.51 ^a^
Cord pH							
Arterial		7.27 (7.24–7.30)			7.28 (7.23–7.31)		0.51 ^a^
Venous		7.32 (7.29–7.35)			7.31 (7.27–7.35)		0.42 ^a^
Respiratory morbidities	10 (10.8)			9 (9.4)			0.75 ^b^
Type of respiratory morbidities							
TTN	9 (9.7)			6 (6.3)			0.30 ^c^
RDS	0 (0)			0 (0)			
Congenital pneumonia	1 (1.1)			3 (3.1)			
Respiratory support							
Nasal oxygen	4 (4.3)			2 (2.1)			0.70 ^c^
cPAP	6 (6.5)			7 (7.3)			
Mechanical ventilation	0 (0)			0 (0)			
NICU admission	2 (2.2)			3 (3.1)			0.63 ^c^
Length of hospital stay (days)			3.51 (2.1)			5.7 (1.5)	0.27 ^d^
Neonatal mortality	0 (0)			0(0)			

^a^ Mann-Whitney test, ^b^ Chi-square test, ^c^ Fisher’s exact test, ^d^ Independent *t*-test.

## Data Availability

Not applicable.

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
