# Peer review of "Antenatal Corticosteroids to Asian Women Prior to Elective Cesarean Section at Early Term and Effects on Neonatal Respiratory Outcomes"

_ijerph, 2022, doi:10.3390/ijerph19095201_

Round 1
Reviewer 1 Report
1.As indicated by the authors, this is a small sample study not powered enough to answer the research question. It can be considered as a 'Pilot study'
2.There is a cochrane review and other studies on the same issue. Hence the study does not add any additional fresh information.
3.Authors mention that the study was blinded. How did they blind the study without a placebo arm?
4.Abstract has some typographical errors including period of gestation as 1266 days
5.First letter of 'Key words' to be capital. It should be 'Material and methods' and not 'Materials'
6.When reference numbers cited are continuous, forst and last number can be put with a dash in between.
7.Why earlier cesarean section alone should cause maternal gastric aspiration, hemorrhage , surgical injuries and anaesthetic complications.
8.Consort flow chart requires correction
9.Why did the author estimate both arterial and venous blood gases?
Author Response
- As indicated by the authors, this is a small sample study not powered enough to answer the research question. It can be considered as a 'Pilot study'
We thank the reviewer for this comment. We agree and have included the words pilot and exploratory study if steroids show significant benefits in our population as recommended (Astecs, Cochrane) and based on more recent trials. (Qatar – Thomas, et al J Perinat Med 2021 May 7;49(7):767 and Iranian - Pirjani etal.BMC Pregnancy and Childbirth (2018) 18:140 ) Our study will encourage future randomised controlled trials to include more diverse ethnic representation on the effects of antenatal corticosteroids for early term.
- There is a cochrane review and other studies on the same issue. Hence the study does not add any additional fresh information.
We acknowledge that there is a Cochrane review on this topic of interest, but having only four trials being included with two of them from the same hospital, as mentioned by the authors of this publication, limits the generalisability of the results. (line 68-69) Furthermore, the trials were predominantly involving the Caucasian population. As stated in our paper, and shown in another previous study (R Patel - Int J Epidemiol 2004 Feb;33(1):107-13.) the earlier onset of labour in Asian and the effects of exposure to antenatal steroids in this diverse ethnic population, we believe is added new information. We have elaborated further on this matter in the introduction and discussion of the revised manuscript. In the light of emerging evidence and caution in the use of antenatal steroids at late preterm (Meta-analysis – Deshmukh and Patole, doi.org/10.1371/journal.pone.0248774 March 22, 2021; and term - - RCOG, published NICE and recommendation no.74), more studies in this area are required in a global context. We believe this paper also fits into the spectrum of the special issue of this journal with the theme - genetic and environmental effects…
- Authors mention that the study was blinded. How did they blind the study without a placebo arm?
We apologise for not being clear on the blinding. Our study is a pragmatic trial that did not involve the use of placebo. Antenatal steroids were not indicated in practice, as such only the trial coordinator, patient and managing obstetrician were aware if antenatal steroids were given. The neonatal team was also not aware of the case in study and the exposure to steroids when called to attend or assess the newborn infant if necessary. We have explained this rather than using the word “blinded” in the revised manuscript.
- Abstract has some typographical errors including period of gestation as 1266 days
Apologies, thank you for picking this up, correction is done for gestation in weeks.
- First letter of 'Key words' to be capital. It should be 'Material and methods' and not 'Materials'
Thank you, correction done. We have replaced the section heading to “Clinical study and methods” instead since this is not a laboratory-based study.
- When reference numbers cited are continuous, first and last number can be put with a dash in between.
Correction done accordingly.
- Why earlier cesarean section alone should cause maternal gastric aspiration, hemorrhage , surgical injuries and anaesthetic complications.
Apologies for not being clear. We have revised to state that in the event that the elective section were to be converted to emergency section. This is in the context of if we were to change our practice to delay planned CS to 39 weeks, our pregnant women might present in labour (as could be the case based on previous reports- R Patel, 2004) prior to the scheduled date which may render a higher number of emergency CS predisposing them to potential maternal gastric content aspiration, haemorrhage, surgical site injuries, and anaesthetic complications as compared to a well-planned elective surgery.
- Consort flow chart requires correction
Correction done.
- Why did the author estimate both arterial and venous blood gases?
This is part of the routine process for all cases of delivery done according to our hospital protocol.
Reviewer 2 Report
Dear Editor,
"Antenatal corticosteroids to Asian women prior to elective caesarean section at early term and effects on neonatal respiratory outcomes" is a good article submitted by Arsad etal.
The manuscript delves in to the utility of antenatal corticosteroid administration and the lack of its use i Asian women undergoing c-section at 37-38 weeks gestation. The article is interesting but the following clarifications may further improve the quality of the same.
- The discussion can be further strengthened by including more details of the ASTECS trial and its relevance to the current study.
- Socioeconomic background, differences in prenatal care ie did the patients receive same level of prenatal or not would clarify the absence of confounding factor.
- Minor English language related errors are present which needs be rectified. For eg: page 141: Criteria is pleural by itself- criterias not needed. Criterion is singular.
- Prolonged hospital stay due to antibiotics. Please clarify. Was this because more babies in that category had suspected pneumonia or was it due to being treated for possible culture negative sepsis.
Author Response
- The discussion can be further strengthened by including more details of the ASTECS trial and its relevance to the current study.
Additional discussion has been added. (line 799-804)
2. Socioeconomic background, differences in prenatal care ie did the patients receive same level of prenatal or not would clarify the absence of confounding factor.
All recruited patients received similar and standard prenatal care in our centre and the ethnic distributions were also similar between groups. Table 1 also showed similarity in the major clinical and demographic characteristics between groups. Unfortunately, we did not look into the subject socio-economic background.
3. Minor English language related errors are present which needs be rectified. For eg: page 141: Criteria is pleural by itself- criterias not needed. Criterion is singular.
Correction done, thank you for pointing out these errors.
4. Prolonged hospital stay due to antibiotics. Please clarify. Was this because more babies in that category had suspected pneumonia or was it due to being treated for possible culture negative sepsis.
It is due to presumptive treatment for pneumonia, and this statement has been included in the revised draft (line 810).
Round 2
Reviewer 1 Report
Nil